# Effects of Cutting Stages and Additives on the Fermentation Quality of Triticale, Rye and Oat Silage in Qinghai-Tibet Plateau

**Jun Ma, Hanling Dai, Hancheng Liu and Wenhua Du \***

Key Laboratory of Grassland Ecosystem of Ministry of Education, Pratacultural Engineering Laboratory of Gansu Province, Sino-U.S. Centers for Grazingland Ecosystem Sustainability, Collage of Pratacultural Science, Gansu Agricultural University, Lanzhou 730070, China
\* Correspondence: duwh@gsau.edu.cn; Tel.: +86-931-7931227

**Abstract:** The Qinghai–Tibet Plateau is sparsely populated and has vast grassland, which plays an important role in the development of animal husbandry. However, during the forage cutting season, frequent rainfall and low temperatures are consistently experienced, which makes it extremely difficult to produce quality hay. The best way to process hay is to produce silages. In this experiment, dry matter yield and silage fermentation quality of dominant annual forages, namely triticale, rye and oat, with Sila-Max and Sila-Mix lactic acid bacteria additives at the five cutting stages, i.e., heading, flowering, grouting, milky and dough stages, were determined. Triticale at the dough stage had the highest dry matter yield among the three forages at the five cutting stages. The optimum harvesting time for triticale, rye and oat to produce quality silage in the Qinghai–Tibet alpine area was the milky stage. Sila-Max lactic acid bacteria additives could significantly improve the fermentation qualities of triticale, rye and oat silages, but the fermentation effect of Sila-Mix on the three silages was not significant. The triticale variety 'Gannong No.2' is the best raw material to produce quality silages in the Qinghai–Tibet alpine area. Overall, quality silage could be made in the Qinghai–Tibet alpine area while using the triticale variety 'Gannong No.2' as the raw material, cutting it at the milky stage and adding Sila-Max as the lactic acid bacteria additive.

**Keywords:** forage species; cutting stages; lactic acid bacteria additives; dry matter yield; silage fermentation quality

## 1. Introduction

The Qinghai–Tibet Plateau is sparsely populated and has vast grassland, which plays an important role in the development of animal husbandry. The Gannan alpine area is located in the northeastern margin of the Qinghai–Tibet Plateau with a cold climate and lack of forage in winter [1]. Oat (*Avena sativa*) is a common forage crop which has been cultivated for a very long time and been popular and welcomed by local farmers in the Qinghai–Tibet Plateau [2]. In recent years, due to the government emphasis on forage planting, rye (*Secale cereale*) has also begun to be widely grown [3]. At the same time, the planting area of triticale (×*Triticosecale* Wittmack) has been expanded year by year due to its advantages of high forage yield and quality, as well as strong lodging resistance [4]. However, during the cutting season of the above forages, frequent heavy rainfall and low temperatures are always experienced; therefore, it is extremely difficult to produce hay of high quality [5]. The best way to process hay is to produce silages, which could effectively preserve the nutritional components of forages and minimize the loss of nutrients from harvesting to storage [6–8]. The silage is soft and juicy with high nutritional value and good palatability [9–11]. Parameters such as pH value, contents of lactic acid (LA), acetic acid (AA), propionic acid (PA), butyric acid (BA), the ratio of ammoniacal nitrogen to the total nitrogen ($NH_3$-N/TN) and content of water-soluble carbohydrate (WSC) are usually used to evaluate the fermentation quality of the silage [12]. LA and AA are produced by lactic acid bacteria and a high content of LA and a suitable content of AA in the silage

could reduce the number of spoilage bacteria, inhibit the growth of spoilage bacteria and improve the fermentation quality of the silage [13]. PA, a kind of short-chain fatty acid, could rapidly reduce the pH value of the silage, prohibit the reproduction of the aerobic bacteria and progress of aerobic glycolysis and prevent the deterioration of the silage [14]. BA is volatile odorous organic compound produced by the decomposition of glucose and lactic acid in the silage, and a high content means a poor quality of the silage [15]. $NH_3$-N/TN reflects the decomposition of protein and amino acid in the silage, and a high ratio means much more protein is decomposed, which negatively affects the fermentation quality of the silage [16]. WSC provides the energy for the fermentation of lactic acid bacteria [17]. Lactic acid bacteria could convert WSC into organic acids (LA and AA) and rapidly reduce the pH of the silage (below 4.6), reduce the contents of PA, BA and $NH_3$-N/TN, and improve fermentation quality of the silage [18]. However, the fermentation quality of the silages had significant differences while cutting at different stages. A large amount of spoilage bacteria such as *Listeria monocytogenes* and *Escherichia coli* was attached to the forage at an early stage, which caused a decline in the fermentation quality [19,20]. If they were cut too late, the dry matter of the forage would be too high, and the fungal infection and heat production would be increased [21] while the carbohydrates content would be decreased [22]. The silage quality of triticale was the best when it was cut at the flowering stage in Temuco, Chile [23], but the best growth stage of triticale for ensiling was the milky stage in Shandong, China [24]. For oat silage, the best cutting time was the boot stage, and oat silage produced at this time contained less starch and non-protein nitrogen and more crude protein in QC, Canada [25], but in Qinghai–Tibet the optimum stage was at the dough stage [26]. The dry matter (DM) content of the raw materials is a very important parameter to judge the harvesting time [27], and 30–35% was the optimal DM content to make silages [28]. Therefore, it is very important to determine the suitable time to cut triticale, rye and oat to produce quality silages. To date, there is no research on the harvesting time of triticale and rye to produce quality silages in the Qinghai–Tibet Plateau. Silage additives also had some influence on the silage fermentation quality, which could increase the nutritional value of the silage, improve the fermentation quality and reduce the loss of storage [29,30]. Compared with non-biological additives such as urea and corn meal, lactic acid bacteria additives had a better effect on the silage, as they not only shortened the fermentation process, but also improved the fermentation quality [31–33]. Lactic acid bacteria additives could reduce the pH value of the silage from 4 to 5 depending on the fodder crop, increase in vitro DM digestibility (IVDMD) and also reduce the contents of the neutral detergent fiber (NDF), AA, PA and $NH_3$-N/TN in the silage [34,35]. The lactic acid bacteria additive lactobacillus rhamanosus-54 improved the anaerobic fermentation profiles by elevating the LA concentration and reducing the AA and BA concentrations in triticale silages [36]. The *Lactobacillus buchneri* additive reduced the DM losses, pH value and AA content, and also improved the aerobic stability of the whole-crop rye silage [37]. The lactic acid bacteria additive Sila-Max could reduce the contents of NDF, acid detergent fiber (ADF), AA, PA, BA, $NH_3$-N/TN and pH, and increase the LA content of the oat silage [26]. Two commonly used commercial lactic acid bacteria additives, namely Sila-Max and Sila-Mix, were widely used in oat and corn silages in China due to their advantages, such as no pollution, moderate price and easy to be used in production [38]. However, no research reports on the use of these two additives in triticale and rye silages in the Qinghai–Tibet alpine area. Therefore, the main questions addressed by this study is (1) the optimum harvesting time for triticale, rye and oat to produce quality silage in the Qinghai–Tibet alpine area, (2) whether the additives Sila-Max and Sila-Mix could improve the silage fermentation qualities of triticale, rye and oat and (3) among the three kinds of forage species, namely triticale, rye and oat, which is the best raw material to produce quality silages in the Qinghai–Tibet alpine area.

## 2. Materials and Methods

### 2.1. Site Description

The field experiment was carried out from September 2017 to August 2018 at the Research Station of Alpine Meadow and Wetland Ecosystems of Lanzhou University (34°55′ N, 102°53′ E, 2950 m above sea level) in Hezuo City, Gansu Province, China. The annual mean temperature was 2.7 °C, precipitation was 550 mm~680 mm, evaporation was 1222 mm and frostless period was 113 d. The depth of the surface runoff is 200 mm~350 mm, subalpine meadow soil dominated this area, and the content of soil organic matter is 13.9%. Rapidly available nitrogen, rapidly available phosphorus and rapidly available potassium in 0~20 cm depth was 248 mg·kg$^{-1}$, 5 mg·kg$^{-1}$ and 198 mg·kg$^{-1}$, respectively, and the pH value was 7.4.

### 2.2. Experimental Materials

Materials used in this experiment were triticale variety Gannong No.2 (referred to as triticale), rye variety Gannong No.1 (referred to as rye) and the popular cultivated local oat variety Minxian (referred to as oat).

The lactic acid bacteria additives were as follows: Sila-Max, produced by Ralco Nutrition Inc., Marshall, MN, USA, containing purified amylase, purified cellulase and lactic acid bacteria, namely *Pediococcus acidilactici*, *Enterococcus faecium* and *Propionibacterium acidipropionici*; and Sila-Mix, which was also produced by Ralco Nutrition Inc., Marshall, MN, USA, and contained lactobacillus *Lactobacillus plantarum*, *Pediococcus acidilactici*, calcium silicate, *Enterococcus faecium*, *Propionibacterium acidipropionici* and *Aspergillus niger*.

### 2.3. Experimental Design and Methods

Split-plot design with three replicates. Forage species were the main plots and included three levels, i.e., triticale (A1), rye (A2) and oat (A3). Cutting stages were the subplots and included five levels, i.e., the heading stage (B1), the flowering stage (B2), the grouting stage (B3), the milky stage (B4) and the dough stage (B5). Lactic acid bacteria additives were the subplots and included three levels, i.e., additive-free treatment (C0), Sila-Max (C1) and Sila-Mix (C2). The area of each plot was 4 m$^2$ (=1 m × 4 m) and there were 135 plots in this experiment. Line seeding was adopted, row spacing was 20 cm and sowing depth was 3~4 cm. Seeding rate was 7.5 million seedlings per hectare and 300 kg/hm$^2$ of nitrogen fertilizer was applied before sowing. Triticale was sown on September 17, 2017, while rye and oats were sown on 18 April 2018, respectively.

Plants were cut at 5 growth stages according to the design. All plants in each plot were cut at approximately 5 cm above the ground, the fresh weight was determined and a sample of about 500 g was randomly collected and firstly dried at 105 °C for 0.5 h, and then dried at 72 °C for 48 h to determine the content of DM in an electro-thermostatic blast oven [39]. Dry matter yield (DMY) was calculated using the following formula (1).

$$DMY = \text{fresh weight} \times DM \text{ content} \tag{1}$$

Fresh samples of about 2 kg were randomly selected from the other materials, chopped into 2 to 3 cm lengths using the fodder chopper and packed into the polyethylene bottle with screw caps (5 L capacity). The amounts of additives were 0.0025 g·kg$^{-1}$ of Sila-Max and 0.5 g·kg$^{-1}$ of Sila-Mix, according to the instructions. The method to produce silage was as follows: additives were dissolved in the distilled water and the control group was only added in an equal amount to the distilled water. Thereafter, additives and water were stirred with silage material uniformly, compacted and sealed. A total of 135 bottles of silage were made in this experiment and fermented for 45 days at the normal temperature. About 500 g of silage sample was taken from top to bottom in 5 different parts of the polyethylene bottle, mixed uniformly to determine the fermentation indices.

### 2.4. Determination of the Fermentation Quality

Twenty grams of silage sample was placed in a triangle bottle with 180 mL of distilled water, sealed with a plastic wrap and extracted at 4 °C for 24 h. Then, it was filtered using four layers of gauze and a piece of filter paper, and the extracted liquid was stored at −20 °C.

The content of water-soluble carbohydrate (WSC) was determined by anthrone–sulfuric acid chromatometry [40], the pH valve was determined by Leici PHS-2F acidimeter (Yidian Scientific Instrument Inc., Shanghai, China), the content of the ammoniacal nitrogen ($NH_3$-N) was determined by phenol–hypochlorite [41] and the ratio of ammoniacal nitrogen to total nitrogen ($NH_3$-N/TN) was calculated.

After the silage's extracted fluid was filtered through 0.22 μm membrane filter, organic acid including the content of lactic acid (LA), acetic acid (AA), propionic acid (PA) and butyric acid (BA) was determined by Agilent 1260 high-performance liquid chromatography (Agilent Technologies Inc., Palo Alto, CA, USA). Chromatographic condition was as follows: SB-AQ $C_{18}$ column (46 mm × 250 mm) was selected, column temperature was 25 °C, the mobile phase was 3% carbinol (0.01 mol/L $(NH_4)_2HPO_4$ = 3:97), the flow rate was 1 mL/min, detective wavelength was 210 nm, injection volume was 20 μL and pH value was 2.70.

### 2.5. Statistical Analysis

The statistical package used for the analysis was SPSS version 19.0. The data analyzed included DM, DMY, WSC, pH valve, $NH_3$-N/TN and organic acid. If significant differences were detected, Duncan's multiple comparison test was undertaken to compare the differences.

## 3. Results

The variance analysis results on the differences in DM, DMY and fermentation quality of silage are shown in Table 1. With the exception of the significant differences in DMY and $NH_3$-N/TN content within the forage species, and no difference in DM content within the forage species, extremely significant differences existed on the other parameters in each variable. Multiple comparisons were needed for the above parameters with significant or extremely significant differences. BA (butyric acid) was not detected in the experiment, so it was not listed in this paper.

**Table 1.** Variance analysis of the differences in DM, DMY and fermentation quality of the silage.

| Variable | DM | DMY | Fermentation Quality | | | | | |
|---|---|---|---|---|---|---|---|---|
| | | | pH | WSC | LA | AA | PA | $NH_3$-N/TN |
| Within the forage species | 0.98 | 3.25 * | 82.91 ** | 1061.06 ** | 136.83 ** | 154.20 ** | 1298.98 ** | 5.35 * |
| Within the cutting stages | 177.23 ** | 17.93 ** | 37.95 ** | 1176.29 ** | 192.12 ** | 277.83 ** | 134.44 ** | 17.04 ** |
| Within the lactic acid bacteria additives | - | - | 385.78 ** | 111.79 ** | 267.30 ** | 389.73 ** | 11.60 ** | 115.1 ** |
| Forage species × Cutting stages | 77.45 ** | 33.68 ** | 45.90 ** | 94.94 ** | 67.28 ** | 80.43 ** | 132.21 ** | 29.73 ** |
| Forage species × Lactic acid bacteria additives | - | - | 11.89 ** | 33.09 ** | 13.20 ** | 20.08 ** | 16.48 ** | 6.75 ** |
| Cutting stages × Lactic acid bacteria additives | - | - | 11.39 ** | 9.74 ** | 11.14 ** | 7.29 ** | 7.70 ** | 6.29 ** |
| Forage species × Cutting stages × Lactic acid bacteria additives | - | - | 5.68 ** | 15.9 ** | 3.67 ** | 11.98 ** | 8.85 ** | 4.37 ** |

Note: * indicates significant differences at the 0.05 level, ** indicates significant differences at the 0.01 level and "-" indicates no interaction. DM: dry matter; DMY: dry matter yield; WSC: water-soluble carbohydrate; LA: lactic acid; AA: acetic acid; PA: propionic acid; $NH_3$-N/TN: the ratio of ammonia nitrogen to total nitrogen.

### 3.1. Differences in DM, DMY and Fermentation Quality of Silage for the Single Factor

3.1.1. Forage Species

The average DMY of A1 at different cutting stages was significantly higher than that of A3, and the average DMY of A1 and A2, as well as A2 and A3 had no significant difference, which indicated that the DMY of A1 was better than that of A3 (Table 2). The average pH value and content of $NH_3$-N/TN of A1 at different cutting stages and additives were

significantly lower than that of A2 and A3, but the average contents of WSC, LA and PA were significantly higher than that of A2 and A3, which indicated that the average fermentation quality of A1 was better than that of A2 and A3. Compared with A3, the average content of WSC and LA of A2 was significantly higher except for the pH value, $NH_3$-N/TN and PA, which indicated that the fermentation quality of A2 was better than that of A3 (Table 2).

**Table 2.** Differences in the average DM, DMY and silage fermentation quality within the forage species, cutting stage and lactic acid bacteria additives.

| Single Factor | Treatment | DM (%) | DMY(t·hm$^{-2}$) | pH Value | WSC(%) | LA(%) | AA(%) | PA(%) | NH$_3$-N/TN(%) |
|---|---|---|---|---|---|---|---|---|---|
| Forage species | A1 | 29.10 ± 4.55 a | 11.97 ± 0.99 a | 4.13 ± 0.04 b | 7.89 ± 0.58 a | 1.97 ± 0.08 a | 0.41 ± 0.03 b | 0.20 ± 0.02 a | 5.39 ± 0.19 b |
|  | A2 | 29.35 ± 4.76 a | 10.48 ± 0.44 ab | 4.35 ± 0.04 a | 4.60 ± 0.68 b | 1.84 ± 0.09 b | 0.44 ± 0.02 a | 0.00 ± 0.00 b | 5.82 ± 0.27 a |
|  | A3 | 28.01 ± 4.29 a | 9.52 ± 0.48 b | 4.34 ± 0.06 a | 2.79 ± 0.31 c | 1.50 ± 0.08 c | 0.32 ± 0.02 c | 0.00 ± 0.00 b | 5.86 ± 0.34 a |
| Cutting stage | B1 | 19.51 ± 0.20 e | 7.31 ± 0.34 d | 4.17 ± 0.04 d | 3.52 ± 0.51 c | 2.14 ± 0.08 a | 0.53 ± 0.03 a | 0.09 ± 0.03 b | 4.70 ± 0.23 c |
|  | B2 | 22.06 ± 0.97 d | 9.48 ± 0.23 c | 4.30 ± 0.07 b | 2.38 ± 0.38 d | 1.92 ± 0.11 b | 0.47 ± 0.03 b | 0.09 ± 0.03 b | 6.02 ± 0.43 a |
|  | B3 | 26.61 ± 0.58 c | 10.59 ± 0.30 bc | 4.25 ± 0.06 c | 3.39 ± 0.60 c | 1.89 ± 0.10 b | 0.34 ± 0.01 c | 0.12 ± 0.03 a | 6.16 ± 0.41 a |
|  | B4 | 30.78 ± 0.78 b | 12.13 ± 0.59 ab | 4.21 ± 0.06 cd | 11.14 ± 0.71 a | 1.74 ± 0.09 c | 0.35 ± 0.02 c | 0.04 ± 0.01 c | 5.56 ± 0.34 b |
|  | B5 | 45.40 ± 0.97 a | 13.76 ± 1.04 a | 4.44 ± 0.07 a | 5.03 ± 0.45 b | 1.15 ± 0.09 d | 0.28 ± 0.02 d | 0.01 ± 0.00 d | 6.01 ± 0.24 a |
| Lactic acid bacteria additive | C0 | - | - | 4.38 ± 0.04 b | 4.77 ± 0.60 b | 1.56 ± 0.08 b | 0.45 ± 0.02 a | 0.06 ± 0.02 b | 6.60 ± 0.28 a |
|  | C1 | - | - | 3.97 ± 0.03 c | 6.04 ± 0.66 a | 2.16 ± 0.07 a | 0.28 ± 0.02 b | 0.06 ± 0.02 b | 4.32 ± 0.21 b |
|  | C2 | - | - | 4.47 ± 0.04 a | 4.46 ± 0.40 c | 1.59 ± 0.08 b | 0.44 ± 0.02 a | 0.08 ± 0.02 a | 6.15 ± 0.19 a |

Note: Different letters in the same column mean significantly differences at $p < 0.05$. '-' indicates no interaction. DM: dry matter; DMY: dry matter yield; WSC: water-soluble carbohydrate; LA: lactic acid; AA: acetic acid; PA: propionic acid; $NH_3$-N/TN: the ratio of ammonia nitrogen to total nitrogen; A1: triticale; A2: rye; A3: oat; B1: heading stage; B2: flowering stage; B3: grouting stage; B4: milky stage; B5: dough stage; C0: additive-free treatment; C1: Sila-Max; C2: Sila-Mix.

### 3.1.2. Cutting Stages

With the delay of the cutting stage, the average DM and DMY of the three forages increased gradually and reached the optimum DM content to produce quality silages at the B4 stage. At the B1 stage, the averaged pH value of the three forage species, with different additives added (the same as follows) was the lowest, followed by B4, and both of them were significantly lower than the other cutting stages. The highest averaged WSC content was observed at the B4 stage, which was significantly higher than the other cutting stages. Along with the growth of forage, the average contents of LA and AA decreased significantly. The average content of PA first increased slowly and significantly decreased afterward, while the B3 stage obtained the highest value, and the B5 stage obtained the lowest content of PA. The average content of $NH_3$-N/TN showed a trend of rising–descending–rising, where the $NH_3$-N/TN of the B1 stage was significantly lower than the other treatments, followed by the B4 stage, and both of them were significantly lower than the other treatments. In terms of DM content and other fermentation parameters, the best cutting stage for the three silages was the B4 stage (Table 2).

### 3.1.3. Lactic Acid Bacteria Additives

For different lactic acid bacteria additive treatments, there were significant differences in the average silage fermentation quality for different forage species cut at different stages (the same as follows). The average pH value, contents of AA and $NH_3$-N/TN of the C1 treatment were significantly lower than those of the C0 and C2 treatments but the average contents of WSC and LA were significantly higher than both of them wherein the average content of WSC was 30% to 35% higher than C0 and C2, and the average content of LA was 35% to 38% higher than C0 and C2. The above analysis showed that the average fermentation quality of C1 was the best, and the average fermentation qualities of C2 and C0 were relatively close and significantly lower than that of C1. BA was not detected in the test, so it was not listed (Table 2).

### 3.2. The Interactions of Two Factors

3.2.1. Forage Species × Cutting Stages

With the delay of the cutting stage, the DM of the three forages increased gradually, and at the B4 stage the DM contents varied from $29.85 \pm 0.68\%$ to $32.34 \pm 1.72\%$, which was the most suitable DM to produce a quality silage [28] (Table 3).

**Table 3.** Differences in the average silage fermentation quality for the interaction of forage species and cutting stage.

| Cutting Stage | Test Materials | DM(%) | pH Value | LA (%) | AA (%) | PA (%) | NH$_3$-N/TN (%) |
|---|---|---|---|---|---|---|---|
| B1 | A1 | $19.77 \pm 0.87$ fg | $4.11 \pm 0.08$ def | $2.03 \pm 0.16$ bc | $0.68 \pm 0.07$ a | $0.27 \pm 0.02$ b | $4.64 \pm 0.38$ def |
| | A2 | $19.12 \pm 0.45$ g | $4.14 \pm 0.06$ cdef | $2.49 \pm 0.06$ a | $0.50 \pm 0.02$ bc | $0.00 \pm 0.00$ d | $4.37 \pm 0.20$ ef |
| | A3 | $19.63 \pm 1.18$ fg | $4.25 \pm 0.07$ cdef | $1.91 \pm 0.10$ c | $0.39 \pm 0.02$ d | $0.01 \pm 0.00$ d | $5.08 \pm 0.53$ cdef |
| B2 | A1 | $22.57 \pm 1.08$ ef | $3.98 \pm 0.05$ f | $2.15 \pm 0.15$ abc | $0.41 \pm 0.03$ cd | $0.27 \pm 0.03$ b | $5.08 \pm 0.19$ cdef |
| | A2 | $23.44 \pm 0.71$ e | $4.39 \pm 0.08$ abcd | $2.33 \pm 0.09$ ab | $0.60 \pm 0.02$ ab | $0.00 \pm 0.00$ d | $5.41 \pm 0.46$ cdef |
| | A3 | $20.18 \pm 1.54$ fg | $4.54 \pm 0.15$ ab | $1.29 \pm 0.09$ d | $0.39 \pm 0.05$ d | $0.00 \pm 0.00$ d | $7.57 \pm 1.05$ ab |
| B3 | A1 | $27.20 \pm 0.05$ cd | $4.09 \pm 0.08$ ef | $2.41 \pm 0.08$ a | $0.32 \pm 0.02$ def | $0.35 \pm 0.03$ a | $5.61 \pm 0.47$ cdef |
| | A2 | $25.46 \pm 1.02$ de | $4.64 \pm 0.06$ a | $1.26 \pm 0.05$ d | $0.38 \pm 0.03$ de | $0.00 \pm 0.00$ d | $8.68 \pm 0.23$ a |
| | A3 | $27.17 \pm 0.40$ cd | $4.01 \pm 0.04$ f | $2.00 \pm 0.06$ bc | $0.32 \pm 0.02$ def | $0.00 \pm 0.00$ d | $4.20 \pm 0.28$ f |
| B4 | A1 | $30.16 \pm 0.56$ bc | $4.05 \pm 0.07$ ef | $1.87 \pm 0.13$ c | $0.27 \pm 0.02$ f | $0.09 \pm 0.01$ c | $5.65 \pm 0.50$ cdef |
| | A2 | $32.34 \pm 1.72$ b | $4.24 \pm 0.08$ cdef | $1.89 \pm 0.08$ c | $0.42 \pm 0.03$ cd | $0.01 \pm 0.00$ d | $4.72 \pm 0.42$ cdef |
| | A3 | $29.85 \pm 0.68$ bc | $4.33 \pm 0.13$ bcde | $1.46 \pm 0.21$ d | $0.35 \pm 0.04$ def | $0.01 \pm 0.00$ d | $6.33 \pm 0.75$ bc |
| B5 | A1 | $45.82 \pm 0.9$ ab | $4.42 \pm 0.12$ abc | $1.40 \pm 0.20$ d | $0.39 \pm 0.04$ d | $0.01 \pm 0.01$ d | $5.97 \pm 0.40$ cde |
| | A2 | $47.16 \pm 0.52$ a | $4.34 \pm 0.08$ bcde | $1.21 \pm 0.09$ d | $0.28 \pm 0.02$ ef | $0.00 \pm 0.00$ d | $5.92 \pm 0.32$ cde |
| | A3 | $43.23 \pm 0.58$ c | $4.57 \pm 0.14$ ab | $0.84 \pm 0.13$ e | $0.17 \pm 0.02$ g | $0.00 \pm 0.00$ d | $6.14 \pm 0.54$ bcd |

Note: Different letters in the same column mean significantly differences at $p < 0.05$. DM: dry matter; LA: lactic acid; AA: acetic acid; PA: propionic acid; NH$_3$-N/TN: the ratio of ammonia nitrogen to the total nitrogen; A1: triticale, A2: rye, A3: oat; B1: heading stage; B2: flowering stage; B3: grouting stage; B4: milky stage; B5: dough stage.

The DMY is shown in Figure 1a. The DMY of the three forage species increased with the growth of the plants and reached the peak at the dough stage. No significant difference was found for the three forage species at the flowering and grouting stages. At the heading stage, A2 had the highest DMY, which was significantly higher than that of A3, and higher than that of A1. At the milky stage (B4), the DMY of A1 ($14.11$ t·hm$^{-2}$) was significantly higher than that of A2 and A3, but no significant differences were found between A2 and A3. At the dough stage, A1 obtained the highest DMY ($17.84$ t·hm$^{-2}$), which was also significantly higher than that of A2 and A3, and there was no significant difference between A2 ($12.04$ t·hm$^{-2}$) and A3 ($11.39$ t·hm$^{-2}$). Evidently, triticale (A1) could harvest the highest DMY at the milky and dough stages when compared to rye (A2) and oat (A3), and the DMY of the three species reached the maximum at the dough stage.

At the B1–B4 cutting stages, the average WSC content of the A1 (triticale) silage with different additives (the same as follows) was significantly higher than that of A2 (rye) and A3 (oat) and peaked at B4 (milky stage) with a value of 14.20% (Figure 1b). For A2, the average WSC content was higher than that of A3, except at the B3 stage. These results demonstrate that A1 (triticale) could provide sufficient fermentation substrate for processing silage, followed by oat.

For the average pH values and NH$_3$-N/TN contents of the three kinds of silages with different additives (the same as follows), no significant differences were found at the B1, B4 and B5 stages. At the B2 stage, the average pH of A1 was significantly lower than that of A2 and A3, and at the B3 stage, the average pH of A1 was significantly lower than that of A2. The average NH$_3$-N/TN content of A1 and A2 was significantly lower than that of A3 at the B2 stage, and the average NH$_3$-N/TN content of A1 was significantly lower than that of A2 at the B3 stage (Table 3).

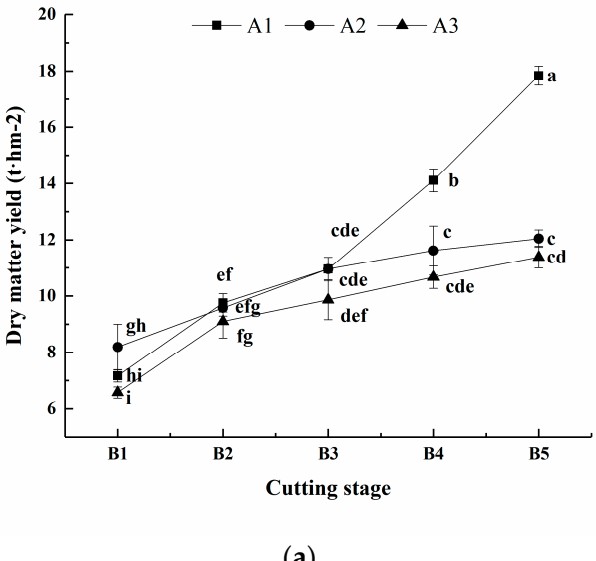

(**a**)

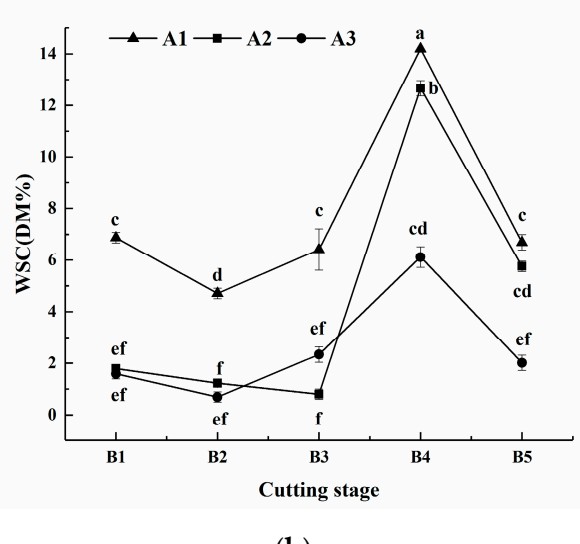

(**b**)

**Figure 1.** The dry matter yield (DMY) (**a**) and average WSC content (**b**) of three forages at different cutting stages. Different letters in each figure mean significantly differences at $p < 0.05$. A1: triticale; A2: rye; A3: oat; B1: heading stage; B2: flowering stage; B3: grouting stage; B4: milky stage; B5: dough stage.

At the B2, B3, B4 and B5 stages, the average LA and PA contents of A1 with different additives (the same as follows) were significantly higher than that of A3. The average PA content of A1 was also significantly higher than that of A3 at the B1 stage but the average LA content of A1 exhibited no significant difference from A3 at the B1 stage. The average PA content of A1 was significantly higher than that of A2 at each stage but only at the B3 stage was the average LA content of A1 significantly higher than that of A2. However, the average LA content of A2 at the B1 stage was significantly higher than that of A1 (Table 3). The above results show that the LA and PA contents in the A1 (triticale) silage could improve the fermentation quality of the silage.

The average AA content of the three silages with different additives (the same as follows) at the B1 and B5 stages was significantly different, as the value of A1 was significantly higher than that of A2 and A3, and the value of A2 was significantly higher than that of A3. However, the average AA content of A1 was significantly lower than that of A2 and A3 at the B4 stage and no significant differences were found between A2 and A3. There was no significant difference in the average AA content of the three silages at the B3 stage (Table 3).

3.2.2. Forage Species × Lactic Acid Bacteria Additives

Differences in the average WSC content for the interaction of forage species and lactic acid bacteria additives are shown in Figure 2. For the same additive treatment, the average WSC content of A1 at different cutting stages (the same as follows) was higher than that of A2 and A3 and the average WSC content of A2 was higher than that of A3. For the same forage species, after adding C1 and C2, the average WSC content was not significantly different from the treatments of C0. This demonstrated that the additives C1 and C2 had no significant influence on the WSC content of triticale, rye and oat silages.

After adding C1, the average pH value, AA content and $NH_3$-N/TN of A1, A2 and A3 at different cutting stages (the same as follows) were significantly decreased compared with no additive treatment (C0). In the meantime, the average LA content increased significantly, and no difference was found regarding the PA content. In contrast, the average pH value, as well as the contents of LA, AA and $NH_3$-N/TN of A1, A2 and A3 with C2 had no significant difference from that of A1, A2 and A3 added with C0. This showed that the additive C1 was better than C2 due to its function of reducing pH value, AA content and $NH_3$-N/TN and increase LA content. For the additive treatment of C1, A1 had the lowest

pH value and highest contents of LA and PA, and its average NH$_3$-N/TN was relatively low. This demonstrated that A1 (triticale) had the best fermentation quality while adding C1 as additive (Table 4).

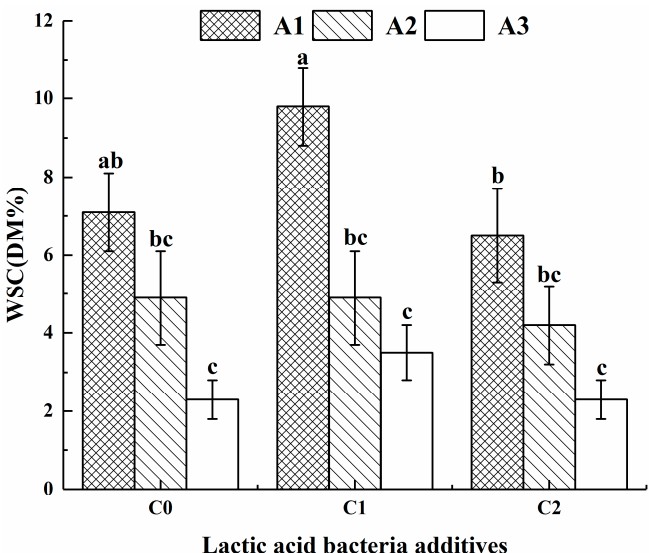

**Figure 2.** Differences in the average WSC content for the interaction between forage species and lactic acid bacteria additives. Different letters mean significant differences at *p* < 0.05. WSC: water-soluble carbohydrate; A1: triticale; A2: rye; A3: oat; C0: additive-free treatment; C1: Sila-Max; C2: Sila-Mix.

**Table 4.** Differences in the average silage fermentation quality for the interaction between forage species and lactic acid bacteria additives.

| Lactic Acid Bacteria Additives | Forage Species | pH Value | LA(%) | AA(%) | PA(%) | NH$_3$-N/TN(%) |
|---|---|---|---|---|---|---|
| C0 | A1 | 4.24 ± 0.06 cd | 1.68 ± 0.13 b | 0.46 ± 0.05 a | 0.18 ± 0.03 b | 6.22 ± 0.26 ab |
| | A2 | 4.37 ± 0.04 bc | 1.71 ± 0.14 b | 0.47 ± 0.04 a | 0.00 ± 0.00 c | 6.24 ± 0.41 ab |
| | A3 | 4.53 ± 0.10 ab | 1.28 ± 0.13 c | 0.42 ± 0.03 ab | 0.00 ± 0.00 c | 7.34 ± 0.66 a |
| C1 | A1 | 3.83 ± 0.02 f | 2.51 ± 0.06 a | 0.28 ± 0.02 cd | 0.17 ± 0.03 b | 4.06 ± 0.12 d |
| | A2 | 4.12 ± 0.06 de | 2.07 ± 0.15 b | 0.35 ± 0.03 bc | 0.01 ± 0.00 c | 4.90 ± 0.55 cd |
| | A3 | 3.97 ± 0.02 ef | 1.91 ± 0.10 b | 0.22 ± 0.02 d | 0.01 ± 0.00 c | 4.00 ± 0.24 d |
| C2 | A1 | 4.33 ± 0.06 c | 1.73 ± 0.11 b | 0.50 ± 0.05 a | 0.24 ± 0.05 a | 5.89 ± 0.22 bc |
| | A2 | 4.55 ± 0.05 a | 1.73 ± 0.15 b | 0.49 ± 0.03 a | 0.00 ± 0.00 c | 6.32 ± 0.37 ab |
| | A3 | 4.52 ± 0.08 ab | 1.30 ± 0.15 c | 0.34 ± 0.03 bc | 0.00 ± 0.00 c | 6.24 ± 0.40 ab |

Note: Different letters in the same column mean significant differences at *p* < 0.05. LA: lactic acid; AA: acetic acid; PA: propionic acid; NH$_3$-N/TN: the ratio of ammonia nitrogen to the total nitrogen; A1: triticale; A2: rye; A3: oat; C0: additive-free treatment; C1: Sila-Max; C2: Sila-Mix.

### 3.2.3. Cutting Stage × Lactic Acid Bacteria Additives

At the same cutting stage, the average WSC content of C1, C2 and C0 of the three forage species (the same as follows) was not significantly different but the average WSC content of C1 was higher than that of C0 and C2. For the same additive, the average WSC content of the B4 stage was significantly higher than that of the other cutting stages (Figure 3). The above results show that the additive C1 had the function of elevating the WSC content in the triticale, rye and oat silages regardless of harvest time, and the three kinds of forage species should be cut at the B4 (milky) stage to obtain more WSC to produce quality silages.

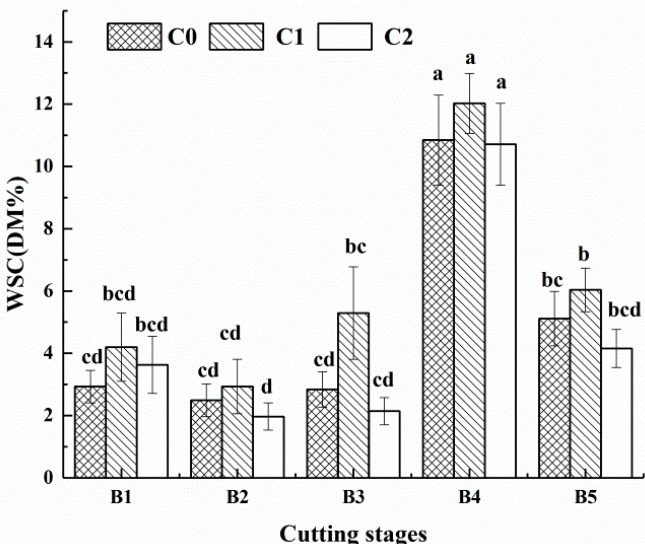

**Figure 3.** Differences in the average WSC content for the interaction between cutting stage and lactic acid bacteria additives. Different letters mean significantly differences at $p < 0.05$. WSC: water-soluble carbohydrate; B1: heading stage; B2: flowering stage; B3: grouting stage; B4: milky stage; B5: dough stage; C0: additive-free treatment; C1: Sila-Max; C2: Sila-Mix.

At the same cutting stage, the C1 treatment had lower average pH values, AA content and $NH_3$-N/TN than that of C0 and C2, and its average LA content was higher than that of C0 and C2; no remarkable difference was found in the average pH value, AA content and $NH_3$-N/TN content of C0 and C2 at the same cutting stage, except for the $NH_3$-N/TN content of C0 and C2 at the B2 stage; and no remarkable difference was found in the average PA content of C0, C1 and C2 (Table 5). This further showed that the additive C1 was beneficial to reduce the pH value, AA content and $NH_3$-N/TN content while elevating the LA content in the triticale, rye and oat silages. For additive C1, lower average pH values, AA content and $NH_3$-N/TN content, as well as higher LA content could be obtained while cutting at the B4 stage.

**Table 5.** Differences in the average fermentation quality for the interaction between cutting stage and lactic acid bacteria additives.

| Cutting Stage | Lactic Acid Bacteria Additives | pH Value | LA (%) | AA (%) | PA (%) | $NH_3$-N/TN(%) |
|---|---|---|---|---|---|---|
| B1 | C0 | 4.25 ± 0.04 cd | 1.87 ± 0.11 bcde | 0.59 ± 0.05 a | 0.09 ± 0.04 ab | 5.70 ± 0.44 bcd |
| | C1 | 3.91 ± 0.03 f | 2.52 ± 0.08 a | 0.38 ± 0.02 c | 0.07 ± 0.03 ab | 3.65 ± 0.09 f |
| | C2 | 4.34 ± 0.02 bc | 2.03 ± 0.12 bcd | 0.61 ± 0.06 a | 0.11 ± 0.06 ab | 4.74 ± 0.18 def |
| B2 | C0 | 4.52 ± 0.15 ab | 1.64 ± 0.19 de | 0.55 ± 0.04 a | 0.08 ± 0.04 ab | 7.79 ± 0.98 a |
| | C1 | 3.98 ± 0.05 f | 2.30 ± 0.18 ab | 0.34 ± 0.05 cde | 0.08 ± 0.03 ab | 4.23 ± 0.18 ef |
| | C2 | 4.41 ± 0.08 bc | 1.83 ± 0.15 cde | 0.50 ± 0.04 ab | 0.12 ± 0.06 ab | 6.04 ± 0.14 bc |
| B3 | C0 | 4.23 ± 0.08 cde | 1.91 ± 0.17 bcde | 0.37 ± 0.01 cd | 0.10 ± 0.05 ab | 6.75 ± 0.65 ab |
| | C1 | 4.08 ± 0.12 def | 2.04 ± 0.20 bcd | 0.26 ± 0.01 def | 0.11 ± 0.05 ab | 5.40 ± 0.88 bcd |
| | C2 | 4.42 ± 0.11 bc | 1.73 ± 0.15 de | 0.40 ± 0.02 c | 0.15 ± 0.07 a | 6.35 ± 0.57 ab |
| B4 | C0 | 4.33 ± 0.08 bc | 1.49 ± 0.10 e | 0.41 ± 0.03 bc | 0.03 ± 0.01 ab | 6.16 ± 0.49 bc |
| | C1 | 3.87 ± 0.02 f | 2.26 ± 0.05 abc | 0.25 ± 0.02 ef | 0.05 ± 0.02 ab | 3.68 ± 0.12 f |
| | C2 | 4.43 ± 0.06 bc | 1.47 ± 0.15 e | 0.38 ± 0.03 c | 0.02 ± 0.01 ab | 6.86 ± 0.44 ab |
| B5 | C0 | 4.57 ± 0.05 ab | 0.88 ± 0.07 f | 0.33 ± 0.04 cde | 0.01 ± 0.01 ab | 6.62 ± 0.18 ab |
| | C1 | 4.02 ± 0.02 ef | 1.70 ± 0.13 de | 0.19 ± 0.03 f | 0.00 ± 0.00 b | 4.66 ± 0.31 def |
| | C2 | 4.74 ± 0.07 a | 0.87 ± 0.10 f | 0.31 ± 0.04 cde | 0.00 ± 0.00 b | 6.75 ± 0.29 ab |

Note: Different letters in the same column mean significantly differences at $p < 0.05$. LA: lactic acid; AA: acetic acid; PA: propionic acid; $NH_3$-N/TN: the ratio of ammonia nitrogen to the total nitrogen; B1: heading stage; B2: flowering stage; B3: grouting stage; B4: milky stage; B5: dough stage; C0: additive-free treatment; C1: Sila-Max; C2: Sila-Mix.

### 3.3. The Interactions of the Three Factors (Forage Species, Cutting Stages and Lactic Acid Bacteria Additives)

#### 3.3.1. WSC Content

Figure 4 showed that, with the same additives at the same cutting stage, the silage produced by A1 had a significantly higher WSC content than that produced by A2 and A3, except for the treatments of B4C0 and B5C2. When the same forage species was cut at the same stage, the WSC contents of A1 and A3 increased significantly or not significantly while adding C1 as additive, as the additives C1 and C2 had no uniform influence on the WSC content in A2. When the same forage species was supplemented with the same additives, the B4 stage obtained significantly higher WSC, followed by the B5 stage. The above results show that A1 (triticale) and A3 (oat) should be harvested at the milky stage to obtain high WSC contents. The additive C1 (Sila-Max) could elevate the WSC content of the silage.

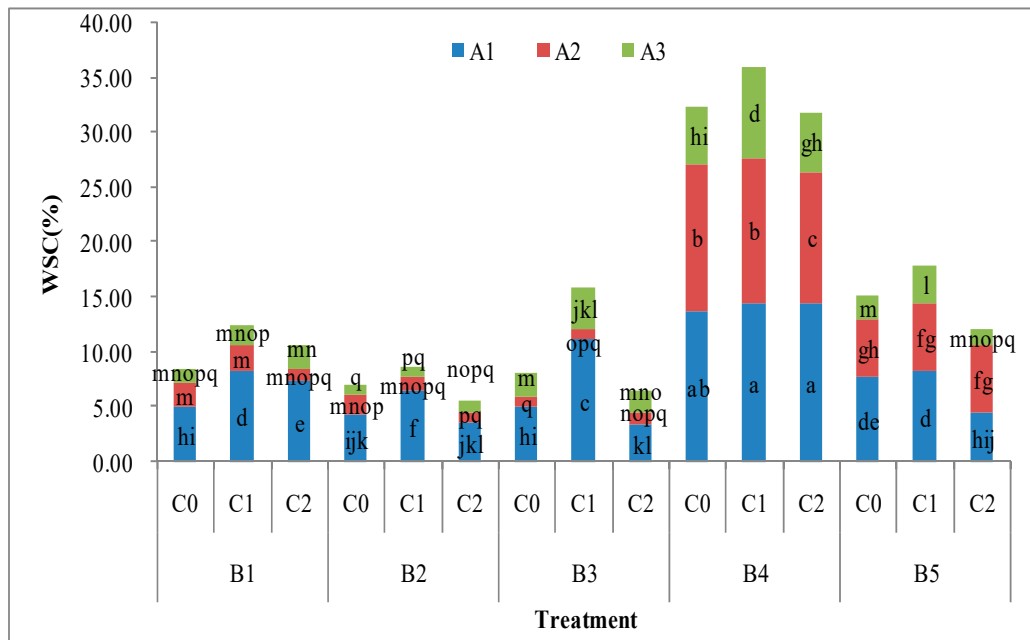

**Figure 4.** Differences in WSC content for the interaction between forage species, cutting stages and additives. Different letters mean significant differences at $p < 0.05$. WSC: water-soluble carbohydrate. A1: triticale; A2: rye; A3: oat; B1: heading stage; B2: flowering stage; B3: grouting stage; B4: milky stage; B5: dough stage; C0: additive-free treatment; C1: Sila-Max; C2: Sila-Mix. a b c d e et al. indicated the significant difference level between treatments ($p < 0.05$). If all marked letters between the two treatments had a same letter, the difference between the two treatments was not significant, and if the marked letters were all different between the two treatments, the difference was significant.

#### 3.3.2. pH Value

With the same additives at the same cutting stage, the pH values of A1 were significantly lower than that of A2, except for B1C0, B5C0 and B5C2, and it was lower than that of A3, except at the B3 stage (Figure 5). When the same forage species was cut at the same stage, the pH values of C1 were significantly lower than that of C0, except for A2B3, and no significant difference was found between C2 and C0, except for A2B1, A3B2, A2B3 and A3B5. When to the same forage species were added the same additives, the B4 stage obtained the lowest values. From the above results, we find that the additive C1 (Sila-Max) could reduce the pH value of triticale, rye and oat silages, and that cutting the above forages at B4 (milky stage) is useful to obtain raw materials with low pH values, and among the three forage species, A1 (triticale) was the best as the silage material in terms of pH value.

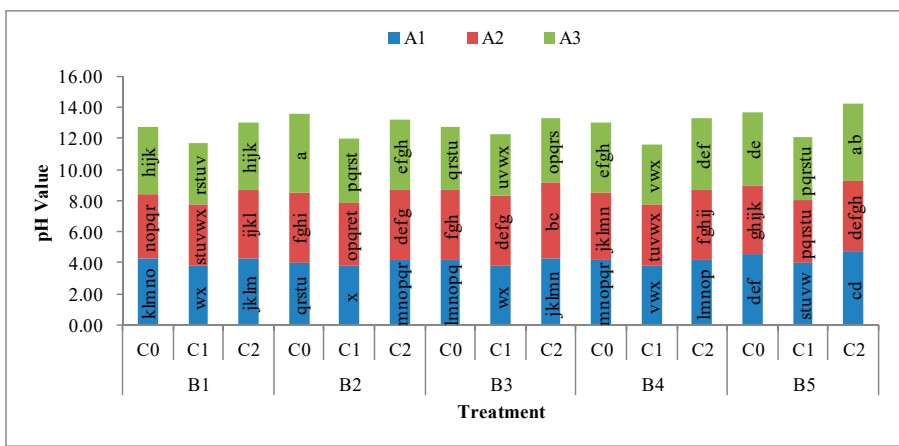

**Figure 5.** Differences in pH value for the interactions of forage species, cutting stage and additive. A1: triticale; A2: rye; A3: oat; B1: heading stage; B2: flowering stage; B3: grouting stage; B4: milky stage; B5: dough stage; C0: additive-free treatment; C1: Sila-Max; C2: Sila-Mix. a b c d e et al. indicated the significant difference level between treatments ($p < 0.05$). If all marked letters between the two treatments had a same letter, the difference between the two treatments was not significant, and if the marked letters were all different between the two treatments, the difference was significant.

### 3.3.3. LA Content

When the three forage species are harvested at the same stage and supplemented with the same additives, the LA content of A1 was significantly higher than that of A3, except for B1C0, B1C2 and B4C1, and it was significantly higher than that of A2 at the B3 stage (Figure 6). When the same forage species was cut at the same stage, the LA content supplemented with C1 was higher than that of C0, and there was no significant difference in the LA content of A1, A2 and A3 with the C2 and C0 additives, except for A3B1, A3B2, A1B3, A1B4 and A3B4. When the same forage species was supplemented with the same additives to make the silage, the suitable cutting stage for A1 should be B2, for A2 it should be B1 and for A3 it should be B4. This showed that silage made from A1 (triticale) had a high content of LA, that the additive C1 (Sila-Max) could elevate the LA content in the silages of triticale, rye and oat, and that harvesting A1 and A2 at an early stage could help obtain a high LA content.

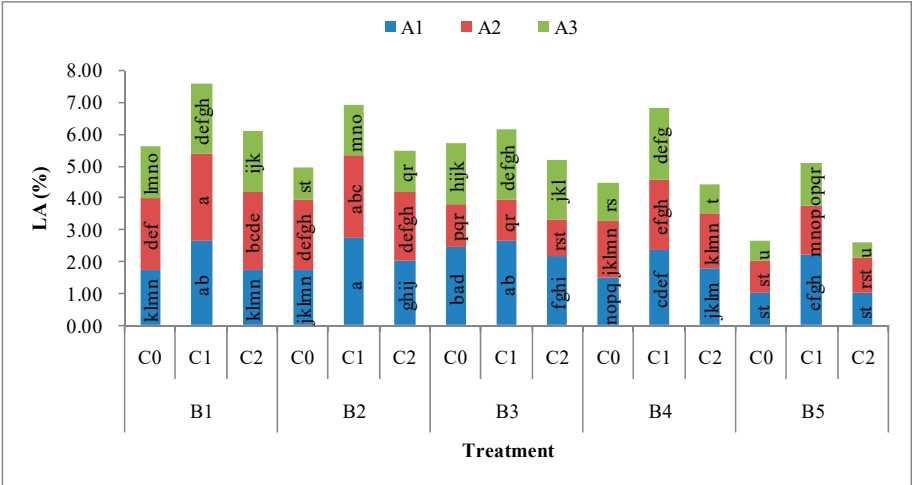

**Figure 6.** Differences in LA content for the interaction between forage species, cutting stage and additive. LA: lactic acid. A1: triticale; A2: rye; A3: oat; B1: heading stage; B2: flowering stage; B3: grouting stage; B4: milky stage; B5: dough stage; C0: additive-free treatment; C1: Sila-Max; C2: Sila-Mix. a b c d e et al. indicated the significant difference level between treatments ($p < 0.05$). If all marked letters between the two treatments had a same letter, the difference between the two treatments was not significant, and if the marked letters were all different between the two treatments, the difference was significant.

### 3.3.4. AA Content

At the same cutting stage and supplemented with the same additive, A1 had a significantly higher AA content at the B1 and B5 stages under the additive treatments C0 and C2, but at the other stages, its AA contents were lower than A2 and higher than A3, and the AA contents of A2 were significantly higher than that of A3 at the B1 and B2 stages (Figure 7). When the same forage species was cut at the same stage, the AA content of the C1 additive was significantly lower than that of the treatments with C0 and C2, except for A2B5, and the AA content of A3B5 was the lowest, which was significantly lower than that of the other treatments; the AA contents of the C2 additive were not significantly different from that of the treatments with C0, except for A1B1, A1B2, A2B5, A3B2 and A3B4. When the same forage species was supplemented with the same additives, they should be cut at the B4 stage to obtain a suitable LA content; otherwise, it may be too elevated. The above analysis demonstrated that harvesting triticale, rye and oat at the B4 (milky) stage could help obtain suitable AA contents and that the additive C1 could reduce the AA content of the silages.

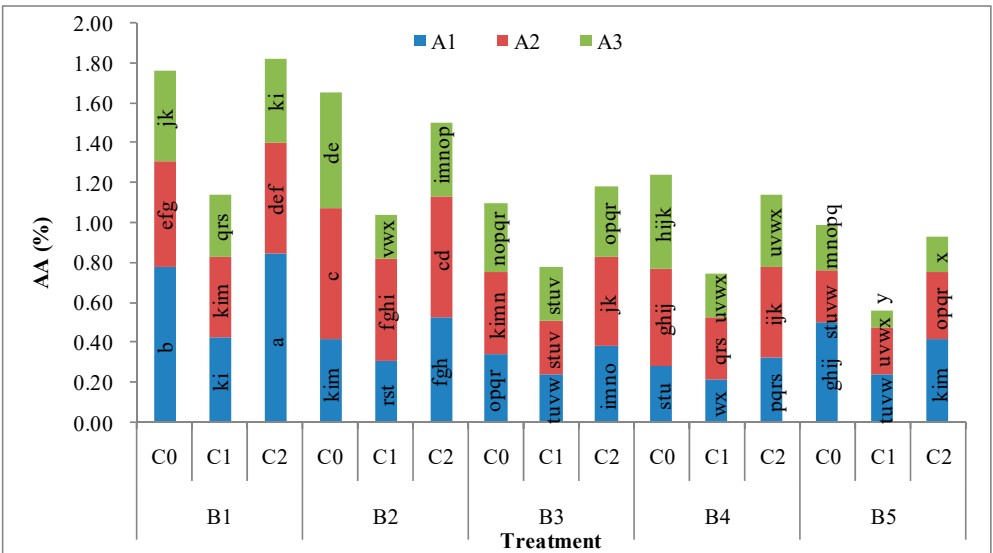

**Figure 7.** Differences in AA for the interaction between forage species, cutting stage and additive. AA: acetic acid. A1: triticale; A2: rye; A3: oat; B1: heading stage; B2: flowering stage; B3: grouting stage; B4: milky stage; B5: dough stage; C0: additive-free treatment; C1: Sila-Max; C2: Sila-Mix. a b c d e et al. indicated the significant difference level between treatments ($p < 0.05$). If all marked letters between the two treatments had a same letter, the difference between the two treatments was not significant, and if the marked letters were all different between the two treatments, the difference was significant.

### 3.3.5. NH3-N/TN Content

The change tendency in $NH_3$-N/TN content was similar to the pH value and AA content (Figure 8). At the same stage and additives, the $NH_3$-N/TN content of A1 was lower than that of A2, except for B1C0, B2C1, B4C0, B4C1, B4C2 and B5C0, and it was lower than that of A3, except for B3C0, B3C1, B3C2 and B5C0. With the exception of B1C1, B3C0, B3C1 and B3C2, the $NH_3$-N/TN content of A2 was lower than that of A3. When the same forage species was cut at the same stage, the additive C1 significantly reduced the $NH_3$-N/TN content, but C2 did not have any significant influence on the $NH_3$-N/TN content. When the same forage species was supplemented with the same additives to make the silage, the suitable cutting stage was B4 for A1 and A2, and B3 for A3. The above analysis demonstrated that the additive C1 (Sila-Max) could reduce the $NH_3$-N/TN content in the silages of triticale, rye and oat. In order to reduce the $NH_3$-N/TN content in silages, A1 (triticale) and A2 (rye) should be harvested at the B4 (milky) stage, and A3 should be harvested at the B3 (grouting) stage. The contents of $NH_3$-N/TN varied depending on the forage species.

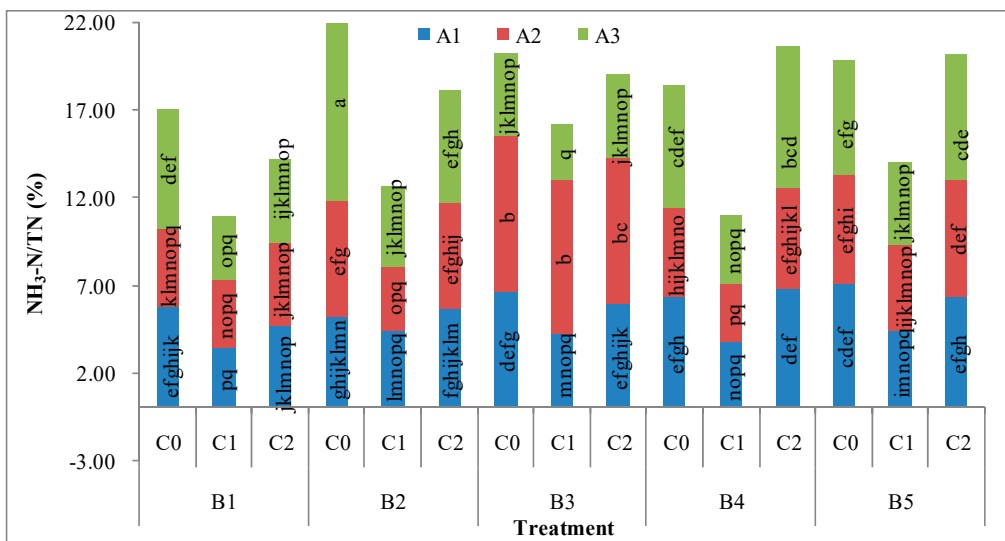

**Figure 8.** Differences in $NH_3$-N/TN contents depending on the interaction between forage species, cutting stage and additive. $NH_3$-N/TN: the ratio of ammonia nitrogen to the total nitrogen. A1: triticale; A2: rye; A3: oat; B1: heading stage; B2: flowering stage; B3: grouting stage; B4: milky stage; B5: dough stage; C0: additive-free treatment; C1: Sila-Max; C2: Sila-Mix. a b c d e et al. indicated the significant difference level between treatments ($p < 0.05$). If all marked letters between the two treatments had a same letter, the difference between the two treatments was not significant, and if the marked letters were all different between the two treatments, the difference was significant.

## 4. Discussion

Forage species, cutting stages and additives had significant effects on the fermentation quality of the silage [23,31]. However, there have been few studies on triticale, rye and oat in the Tibetan Plateau. By studying the effects of cutting stages and additives on the fermentation quality of three main forage grasses in this region, it may provide a basis for the selection of forage species, cutting stage and additive for producing high-quality silages in the Qinghai–Tibet Plateau.

### 4.1. The Optimum Harvesting Time for Triticale, Rye and Oat to Produce Quality Silage in Qinghai–Tibet Alpine Area Was the Milky Stage

The DM content in the raw material is an important parameter to make silages. If the DM content in the raw material is insufficient and the moisture content is too high (more than 85%), lactic acid bacteria cannot multiply rapidly, butyric acid bacteria and other bacteria proliferate and form a dominant flora [42]. The butyric acid bacteria make the protein corrupted by dehydrogenation, decarboxylation and redox, which promote the growth of acid-intolerant putrefactive microorganisms, and make the silage deteriorated, smelly and viscous; then, glucose and lactic acid decompose to produce volatile odor butyric acid [43]. On the other hand, if the DM content in the raw material is too high and the moisture content is insufficient, it is difficult to be compacted during silage production, resulting in unexpected air, meaning that a large number of aerobic bacteria will multiply and also make the silage rot and deteriorate [44]. Therefore, a suitable DM content should be contained in the raw materials to make quality silage and 30% to 35% of DM is preferred [28]. In this study, the DM contents of triticale, rye and oat harvested at the milky stage varied from 29.85% to 32.34% (Table 3), which makes it possible to make quality silage. Additionally, at this stage, the WSC contents in the three kinds of silages were all significantly higher than that of the other stages (Figure 1b), which provided suitable conditions and energy for the fermentation of lactic acid bacteria [45]. The fermentation results show that triticale, rye and oat silages made at the milky stage had better fermentation quality because they had low pH values (4.05–4.33), contents of AA (0.27–0.42%), PA (0.01–0.09%) and NH3-N/TN (4.72–6.33%) and high values in LA content (1.46–1.87%) (Table 3). In contrast, the pH value

and NH3-N/TN content of the three kinds of silages made at the dough stage were high and the LA content was low (Table 3). The main reason is for this is that the WSC in the raw material is synthesized into starch at the dough stage and WSC content is sharply dropped [45], which restricts the reproduction of lactic acid bacteria, affects the reduction in pH value and causes the multiplication of miscellaneous bacteria to affect the fermentation quality [46].

### 4.2. Lactic Acid Bacteria Additives Sila-Max Could Improve the Silage Fermentation Qualities of Triticale, Rye and Oat

Lactic acid bacterial inoculants could use lactic acid bacteria to ferment the substrate glucose to produce lactic acid, so it could increase the LA content and reduce the pH value and $NH_3$-N/TN of the maize silage [47]. Lactic acid bacteria and molasses, significantly elevated the concentration of LA and decreased the concentration of AA, PA, BA and ammonia-N in the cassava foliage silage [48]. This study obtained similar results, showing that the lactic acid bacterial additive Sila-Max significantly increased the contents of WSC and LA and reduced the pH and contents of AA and $NH_3$-N/TN in triticale, rye and oat silages, which was consistent with Zhao et al. (2018), Meeske and Basson (1998) and Li et al. (2021) [26,47,48]. The main reason for this is that Sila-Max requires a very short amount of time to enter the stable period of fermentation to produce a large amount of lactic acid to rapidly reduce the pH and inhibit the production of BA and ammonia-N [26,38]. Therefore, the lactic acid bacteria additive Sila-Max could improve the silage fermentation qualities of triticale, rye and oat silages in the Qinghai–Tibetan Plateau. However, for Sila-Mix, it elevated the pH value and PA content, reduced the WSC content and did not have any influence on the contents of LA and AA in triticale, rye and oat silages, which is in accordance with Chen et al. (2021), showing that the inoculation of exogenous lactic acid bacteria exerted a limited influence on the silage fermentation and bacterial community compositions of reed canary grass straw on the Qinghai–Tibetan Plateau [49].

### 4.3. Triticale Variety 'Gannong No.2' Is the Best Raw Material to Produce Quality Silage in Qinghai–Tibet Alpine Area

DMY is the weight while the DM content of the forage is 100%, which plays an important role for silage production [50]. High DMY means high economic benefits [51]. The DMY of the triticale variety 'Gannong No.2' cut at the milky stage (14.11 t·hm−2) was significantly higher than that of rye and oat (Figure 1a) due to its thick stem (4.25 mm) and blade (317.50 μm), abundant tillers (5–6/plant) and large leaf size (20.34 cm2) [52]. In contrast, although the plant height of the rye variety 'Gannong No.1' is the highest, due to its slender stems (3.21 mm), small size of blades (15.22 cm2) and thin leaves (243.00 μm) [52], it obtained a low DMY. Additionally, for oat, the lowest height decided its low DMY [53]. The DMY results obtained from this experiment are consistent with the results of Zhao et al. (2018) and Wang et al. (2020) [26,54]. This demonstrated that the triticale variety 'Gannong No.2' is much more economic than that of oat and rye to make quality silages in the Qinghai–Tibet Plateau.

WSC is the most important energy substance for maintaining the fermentation quality of the silage, and high contents of WSC could provide sufficient energy substances for the silage [55]. The WSC in the silage should be at least 1% ~ 1.5% of the fresh weight to ensure successful fermentation [56]. In this experiment, the WSC content of the triticale silage produced at the milky stage was 13.73% (with no additive), indicating that triticale itself contained higher WSC. Researchers also mixed triticale in some leguminous silages to elevate the WSC content of the silage [57]. As an important volatile fatty acid in silage, PA possesses antifungal properties and it plays an important role in inhibiting the aerobic spoilage of the silage to preserve its nutritional content [58,59]. In the oat silage, PA effectively inhibited the formation of undesirable microorganisms and BA, enhanced the aerobic stability and improved the fermentation quality [60]. In this study, the PA content in the triticale silage was about 10–20 times higher than that of rye and oat, showing that the triticale silage had good aerobic stability, which can be preserved longer than that of

rye and oat. Moreover, the pH value and NH3-N/TN content were also low in the triticale silage. Therefore, the triticale variety 'Gannong No.2' was the best raw material to produce a quality silage in the Qinghai–Tibet alpine area.

## 5. Conclusions

The optimum harvesting time for triticale, rye and oat to produce quality silages in the Qinghai–Tibet alpine area was the milky stage. The lactic acid bacteria additive Sila-Max could significantly improve the silage fermentation qualities of triticale, rye and oat, but the fermentation effect of Sila-Mix on the three silages was not significant. The triticale variety 'Gannong No.2' is the best raw material to produce quality silages in the Qinghai–Tibet alpine area. Overall, a quality silage could be made in the Qinghai–Tibet alpine area by using the triticale variety 'Gannong No.2' as the raw material, cutting it at the milky stage and adding Sila-Max as the lactic acid bacteria additive.

**Author Contributions:** J.M. conceived and designed the experiments, analyzed the data, prepared figures and/or tables, authored and approved the final draft. H.D. performed the experiments, reviewed drafts of the paper and approved the final draft. H.L. reviewed and approved the final draft. W.D. conceived and designed the experiments, authored or reviewed drafts of the paper and approved the final draft. All authors have read and agreed to the published version of the manuscript.

**Funding:** This study was supported by National Natural Science Foundation of China (32260339), Key Research and Development Projects in Gansu Province (20YF8NA129), Major Science and Technology project of Tibet (XZ202101ZD003N) and Higher Education Industry Support Program Project of Gansu Province (2022CYZC-49). The funders had no role in study design, data collection and analysis, decision to publish or preparation of the manuscript.

**Data Availability Statement:** All datasets are included in the manuscript, and additional datasets are available upon reasonable request.

**Acknowledgments:** The authors gratefully acknowledge National Natural Science Foundation of China (32260339), Key Research and Development Projects in Gansu Province (20YF8NA129), Major Science and Technology project of Tibet (XZ202101ZD003N) and Higher Education Industry Support Program Project of Gansu Province (2022CYZC-49).

**Conflicts of Interest:** The authors declare that they do not have any conflict of interest.

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
