# Peer review of "Effects of Cutting Stages and Additives on the Fermentation Quality of Triticale, Rye and Oat Silage in Qinghai-Tibet Plateau"

_agronomy, doi:10.3390/agronomy12123113_

Round 1
Reviewer 1 Report
Dear Authors,
The obtained by the authors results are interesting, but in my opinion, the reviewed manuscript needs many improvements and clarifications. However, there are some aspects that are not sufficient to justify and discuss the results obtained.
1. The research question is one of the most important parts of your research. What is the main question addressed by the research?
2. What is the justification for your research on cutting stages and additives in the fermentation and nutritive quality of forages? Why you have conducted your study in the first place? This part of your paper needs to explain the uniqueness and importance of your research.
3. The review of the literature is not thorough so the reader is not given an adequate background about the topic. Authors can improve the introduction part and discuss it. There is not enough information on the nutritive value and fermentation of forage species.
4. When you are using abbreviations in tables and figures, all abbreviations used in tables and figures should be defined in the table note or figure caption, respectively, even though the abbreviations will also be defined in the text if they are used there.
5. There are also many repetitions in the discussions and in any case, they must be rewritten.
6. Are the conclusions consistent with the evidence and arguments presented? Do they address the main question posed? The arguments and data presented are not complete in establishing the core idea presented.
7. The English in the present manuscript is not of publication quality and requires improvement. Please carefully proofread spell check to eliminate grammatical errors.
In my review, I identified some major weaknesses of the manuscript, mostly linked to the introduction, methodological aspects in the experimental design, and presentation of the results. Therefore, as the manuscript is valuable and brings interesting results, I would encourage the authors to carefully revise the manuscript taking into account my suggestions and remarks, in order to definitively improve the quality of the work.
Author Response
Reviewer
Major comments
- The research question is one of the most important parts of your research. What is the main question addressed by the research?
Many thanks for your comment! Qinghai-Tibet Plateau is a very important area to cultivate annual forages to development the animal livestock industry. However, during the forage harvesting season, it’s always with frequent heavy rainfall and low temperature, therefore, it is extremely difficult to produce quality hay. The best way to process them is to produce silages. The main question addressed by the research is 1) the optimum harvesting time for triticale, rye, and oat to produce quality silage in Qinghai-Tibet alpine area, 2) whether additives Sila-Max and Sila-Mix could improve the silage fermentation qualities of triticale, rye and oat, and 3) Among three kinds of forage species, triticale, rye, and oat, which one is the best to produce quality silage in Qinghai-Tibet alpine area. I have supplement them at the end of the introduction. Please refer to them.
- What is the justification for your research on cutting stages and additives in the fermentation and nutritive quality of forages? Why you have conducted your study in the first place? This part of your paper needs to explain the uniqueness and importance of your research.
Thanks for your suggestion! The best time to cut forage to make quality silage is the dry matter (DM) content of the raw materials and 30%-35% is the optimal DM content to produce quality silage (Wang and Wang, 2003). Because there is not this kind of basic information by now, so in this manuscript, we designed 5 growth stages of triticale, rye, and oat to cut them to find the suitable DM content to make quality silage. And for each kind of forage we cut it at the growth stages of heading, flowering, grout, milky, and dough, respectively. Please refer to line 58-71 in the manuscript.
There is no justification on additives in this study and we selected two widely used commercial lactic acid bacteria additives (Sila-Max and Sila-Mix) in oats and corn silage in China in this study. I have supplemented these contents in the manuscript and marked them in red. Please refer to line 84-90 in the manuscript.
The uniqueness and importance of the research: Qinghai-Tibet Plateau is sparsely populated and has vast grassland to develop the animal husbandry, which plays an important role in the development of animal husbandry. However, during the cutting season of the dominated annual forage species (triticale, rye, and oat), it’s always with frequent heavy rainfall and low temperature, and it is extremely difficult to produce quality hay. The best way to process them is to produce silages, which could effectively preserve the nutritional components of forages and minimize the loss of nutrients from harvesting to storage. To date, there was no research on the harvesting time and additives of triticale and rye to produce quality silages in Qinghai-Tibet Plateau. I have added these contents in the introduction and marked them in red. Please check them.
- The review of the literature is not thorough so the reader is not given an adequate background about the topic. Authors can improve the introduction part and discuss it. There is not enough information on the nutritive value and fermentation of forage species.
Many thanks for your suggestion! I also realized this problem. I have adjusted the writing order of cutting stages and additives in the introduction and added the following contents in the introduction: the parameters used to evaluate the fermentation quality of silages and their functions in the fermentation, the relationship between each parameter and fermentation quality, the optimal DM content to make silage, and the the main question addressed by the research. I have marked them in red and please refer to line 42-58.
There are lots of information on the nutritive value of hays of the three forage species, triticale, rye, and oat, in other areas, and the nutrition was as follows: triticale (CP content: 7% - 13%, NDF: 48%-56%, ADF: 29%-35%), rye (CP: 6%-11%, NDF: 52%-62%, ADF: 35%-42%), Oat (CP: 6%-10%, NDF: 47%-57%, ADF: 32%39%). But information on the nutritive value and fermentation of silages of the three forage species was limited. I have supplemented them in the introduction. Please refer to line 79-89. Because nutritional quality did not involved in this manuscript, the basic information on the nutrition quality was not listed.
- When you are using abbreviations in tables and figures, all abbreviations used in tables and figures should be defined in the table note or figure caption, respectively, even though the abbreviations will also be defined in the text if they are used there.
Thanks for your suggestion! I have redefined all abbreviations used in tables and figures in the table note or figure caption, respectively. Please refer to line 171-174, 203-207, 227-230, 246-248, 287-289, 302-304, 315-318, 331-334, 349-352, 365-367, 383-385, 402-404, 421-423 in the manuscript.
- There are also many repetitions in the discussions and in any case, they must be rewritten.
Thanks for your suggestion! I have rewritten the discussion of the manuscript and there is not any repetitions existed. Please refer to line 425-504 in the manuscript.
- Are the conclusions consistent with the evidence and arguments presented? Do they address the main question posed? The arguments and data presented are not complete in establishing the core idea presented.
Very sorry for this! In the previous version, I ignored that dry matter content in the raw material is the most important parameter to determine the harvesting time to make silage and 30%-35% was the optimal dry matter content to make silage [28]. So I used the grey correlation analysis method to comprehensively evaluate the silage fermentation quality of forages added with C1 and cutting at different growth stages. Although the similar results got from the comprehensive evaluation, it is not correct. In this version, firstly, I carefully checked the raw data of the dry matter content and corrected the wrong data previously input because of carelessness. Then I determined the optimal cutting time-milky stage mainly based on the dry matter content, of course, other fermentation parameters also considered. Because at this stage, the WSC contents in three kinds of silages were all significantly higher than that of the other stages (Figure 1(b)). Fermentation results really showed that triticale, rye, and oat silages made at the milky stage had better fermentation quality because they had low values in pH (4.05-4.33), contents of AA (0.27%-0.42%), PA (0.01%-0.09%), and NH3-N/TN (4.72%-6.33%), and high values in LA content (1.46%-1.87%) (Table 3). After this, I determined the best additive according to the fermentation quality of silages at the milky stage. At the end, I determined the optimal forage species to make quality silage in Qinghai-Tibet alpine area. I also discussed the above results in the discussion part and marked them in red. Please check them.
- The English in the present manuscript is not of publication quality and requires improvement. Please carefully proofread spell check to eliminate grammatical errors.
Thanks for your comments! The manuscript had been edited by Wiley Editing Services before submitting to the journal last time. For the revised version, I have asked a native English-speaking colleague to edit the language.

Reviewer 2 Report
The article contains a serious and complete study of the preparation of silage in an area unique in its geographical location. The influence of the addition of commercial preparations containing lactic acid bacteria on the content of certain acids in the silage and the reduction of its natural toxicity due to the unavoidable content of ammonia nitrogen was investigated.
While I have no objections to the design of the study and the unequivocal results obtained, I do have several remarks about the layout of the text and missing data.
The introduction mentions the role of lactic acid bacteria as antagonists of pathogenic bacteria and fungi in silages. References from recent years that are right on target and issued by MDPI can be cited here, for example, https://doi.org/10.3390/nu14102038.
"Materials and methods": please, wherever apparatus used or commercial manufacturers are listed, they should be listed with their full names, state, and country. For example, "American Ralco Nutrition Company" should be listed as "Ralco Nutrition Inc., MN, USA", Agilent Technologies Inc., and so on.
Line 84: propionibacterium to be capitalized. Lines 83-84: the part "and the fermentation productions" is redundant if the following types of bacteria are present in the preparation.
The descriptions of the figures should be more complete, for example, in Figure 1 it should be indicated what A1, A2, A3, etc. mean. In the text "fig." should be replaced by "figure".
Lines 183, 194, 212, 263, 293: are these sub-headings? They can be given a number such as 3.2.1; 3.3.1, etc.
In the conclusions, the abbreviation "TOPSIS" occurs for the first time. It should be introduced much earlier and the abbreviation - described.
References: after each reference, a link to the article should be indicated (http://doi.org)
In conclusion, the article should be published as it contains interesting and new data, but with the necessary improvements in the presentation of the material.
Author Response
Reviewer
Major comments
- The introduction mentions the role of lactic acid bacteria as antagonists of pathogenic bacteria and fungi in silages. References from recent years that are right on target and issued by MDPI can be cited here, for example, https://doi.org/10.3390/nu14102038.
Many thanks for your comments! I have cited recent references published by MDPI on the role of lactic acid bacteria as pathogenic and fungal antagonists in silage. Please find them in line 598-601 in the manuscript.
2."Materials and methods": please, wherever apparatus used or commercial manufacturers are listed, they should be listed with their full names, state, and country. For example, "American Ralco Nutrition Company" should be listed as "Ralco Nutrition Inc., MN, USA", Agilent Technologies Inc., and so on.
Thanks for your comments! I have revised them in the manuscript and please find it in line 110, 113, 148-149, 154-155 in the manuscript.
3.Line 84: propionibacterium to be capitalized. Lines 83-84: the part "and the fermentation productions" is redundant if the following types of bacteria are present in the preparation.
Many thanks for your comments! I have capitalized the word "Propionibacterium" and removed the part "and the fermentation productions". Please check it in line 112 and 111 in the manuscript.
4.The descriptions of the figures should be more complete, for example, in Figure 1 it should be indicated what A1, A2, A3, etc. mean. In the text "fig." should be replaced by "figure"
Many thanks for your comments! I have completed all the descriptions of the figures and replaced "fig." by "figure". Please refer to lines 171-174, 203-207, 227-230, 246-248, 287-289, 302-304, 315-318, 331-334, 349-352, 365-367, 383-385, 402-404, 421-423, 233 in the manuscript.
- Lines 183, 194, 212, 263, 293: are these sub-headings? They can be given a number such as 3.2.1; 3.3.1, etc.
Many thanks for your comments! I have given them the numbers, such as "3.1.1; 3.1.2; 3.1.3; 3.2.1; 3.2.2; 3.2.3" and regarded them as sub-headings. Please refer to lines 176, 186, 209, 220, 277, and 305 in the manuscript.
- In the conclusions, the abbreviation "TOPSIS" occurs for the first time. It should be introduced much earlier and the abbreviation - described.
Many thanks for your comments! I have deleted them in the revised manuscript.
- References: after each reference, a link to the article should be indicated (http://doi.org)
Many thanks for your comments! I have revised them according to the format of the journal. Please check it in line 559-668 in the manuscript.

Round 2
Reviewer 1 Report
Dear authors :
Where you kept your silage samples? and which temperature?
Please provide the Gas Chromatography Graph with the sampling pick and time for each treatment.
Discussions should be improved and answer your research questions and objectives.
Author Response
Collage of Pratacultural science, Gansu Agricultural University,
Gansu Province, Lanzhou 730070, China
(sanxingbaoxi2000@163.com)
December 1st, 2022
Dear Editors and Reviewers,
Thank you very much for your careful review of our manuscript titled “Effects of cutting stages and additives on the fermentation quality of triticale, rye and oat silage in Qinghai-Tibet Plateau”. We do appreciate the supportive comments. We have now completed the revision which included several supplemental contents and the answer of the question from our submitted draft.
As directed, we have now prepared a revised version of our manuscript, and we have also included a full point-by-point response to the Reviewer's comments. Many thanks for the ongoing support of our manuscript.
Respectfully,
Jun Ma
Collage of Pratacultural Science
Gansu Agricultural University.
On behalf of all authors.
Reviewer
Major comments
1.Where you kept your silage samples? and which temperature?
Many thanks for your comment! When triticale, rye and oat reached the heading, flowering, grouting, milky and dough stage, respectively, we immediately harvested them and made fermentation according to the design. Then the silages were placed in our laboratory and fermented for 45 days at the normal room temperature. Once the fermentation days of each treatment reached 45 days, we immediately took silage samples from each treatment to determine the fermentation parameters. Silage samples and extracts were then stored in a freezer at 4℃ for use.
2.Please provide the Gas Chromatography Graph with the sampling pick and time for each treatment.
Many thanks for your comment! In our study, we used Agilent 1260 high-performance liquid chromatography (Agilent Technologies Inc., CA, USA) to determine the contents of lactic acid, acetic acid, propionic acid, butyric acid in 2018. The graphs were kept on my laptop, but my laptop was broken last year so the image was lost. Today I gave the lab administrator a call and wanted to copy the data from the Agilent high-performance liquid chromatography's computer, but he told me that the liquid chromatography was upgraded at the end of 2018 and the data before 2018 was lost. So very sorry that I could not provide the graphs at this moment.
The time to pick the sample was as follows, heading stage: June 25th, flowering stage: July 12th, grouting stage: July 22nd, milky stage: August 7th, and the dough stage: September 5th.
- Discussions should be improved and answer your research questions and objectives.
Many thanks for your comment! In the revised version of our manuscript, we supplemented the main question addressed by the research according to the comments of reviewer, which was 1) the optimum harvesting time for triticale, rye, and oat to produce quality silage in Qinghai-Tibet alpine area, 2) whether additives Sila-Max and Sila-Mix could improve the silage fermentation qualities of triticale, rye and oat, and 3) Among three kinds of forage species, triticale, rye, and oat, which is the best raw material to produce quality silage in Qinghai-Tibet alpine area. Please refer to the end of the introduction. In the discussion, in order to answer the question addressed in the introduction, we did a major revision. Please refer to lines 426-505. Then we got the conclusion and we think to answer the question and accomplish the objective. Please refer to lines 506-514.
The conclusion was as follows,
the optimum harvesting time for triticale, rye, and oat to produce quality silage in Qinghai-Tibet alpine area was the milky stage. Lactic acid bacteria additives Sila-Max could significantly improve the fermentation qualities of triticale, rye, and oat silages, but the fermentation effect of Sila-Mix on three silages was not significant. Triticale variety ‘Gannong No.2’ is the best raw material to produce quality silage in Qinghai-Tibet alpine area. Overall, quality silage could be made in Qinghai-Tibet alpine area while using triticale variety ‘Gannong No.2’ as the raw material, cutting it at the milky stage, and adding Sila-Max as the lactic acid bacteria additive.
The discussion was as follows,
Forage species, cutting stages and additives had significant effects on fermentation quality of silage [23, 31]. However, there had been few studies on triticale, rye and oat in the Tibetan Plateau. Through studying the effects of cutting stages and additives on the fermentaion quality of three main forage grasses in this region, it would provide a basis for the selection of forage species, cutting stage and additives for producing high-quality silage in the Qinghai-Tibet Plateau.
4.1 The optimum harvesting time for triticale, rye, and oat to produce quality silage in Qinghai-Tibet alpine area was the milky stage
DM content in the raw material is an important parameter to make silage. If the DM content in the raw materials insufficient and the moisture content is too high (more than 85%), lactic acid bacteria cannot multiply rapidly, butyric acid bacteria and other bacteria proliferate will form a dominant flora [42].The butyric acid bacteria make the protein corrupted by dehydrogenation, decarboxylation and redox, which promote the growth of acid-intolerant putrefactive microorganisms, and make silage deteriorated, smelly and viscous, then glucose and lactic acid decompose to produce volatile odor butyric acid [43].On the other hand, if the DM content in the raw material is too high and the moisture content is insufficient, it is difficult to be compacted during silage resulting in unexpected air, a large number of aerobic bacteria will multiply and also make silage to rot and deteriorate [44]. Therefore, suitable DM content should be contained in the raw materials to make quality silage and 30% to 35% of DM is preferred [28]. In this study, DM contents of triticale, rye, and oat harvested at the milky stage varied from 29.85% to 32.34% (Table 3), which is possible to make quality silage. And at this stage, the WSC contents in three kinds of silages were all significantly higher than that of the other stages (Figure 1(b)), which provided suitable condition and energy for the fermentation of lactic acid bacteria [45]. Fermentation results really showed that triticale, rye, and oat silages made at the milky stage had better fermentation quality because they had low values in pH (4.05-4.33), contents of AA (0.27%-0.42%), PA (0.01%-0.09%), and NH3-N/TN (4.72%-6.33%), and high values in LA content (1.46%-1.87%) (Table 3). In contrast, the pH value and NH3-N/TN content of three kinds of silages made at the dough stage was high and the LA content was low (Table 3). The main reason is that, the WSC in the raw material is synthesized into starch at the dough stage and WSC content sharply dropped [45], which restricted the reproduction of lactic acid bacteria, affected the reduction of pH value, and caused the multiplication of miscellaneous bacteria to affect the fermentation quality [46].
4.2 Lactic acid bacteria additives Sila-Max could improve the silage fermentation qualities of triticale, rye, and oat
Lactic acid bacterial inoculants could use lactic acid bacteria to ferment the substrate glucose to produce lactic acid, so it could increase the LA content and reduce pH value and NH3-N/TN of maize silage [47]. Lactic acid bacteria, molasses, significantly elevated the concentration of LA and decreased the concentration of AA, PA, BA and ammonia-N in cassava foliage silage [48]. This study got the similar results showing that lactic acid bacterial additive Sila-Max significantly increased contents of WSC and LA and reduced the pH and contents of AA and NH3-N/TN in triticale, rye, and oat silages, which was consistent with Zhao et al.(2018), Meeske and Basson (1998) and Li et al. (2021)[26, 47, 48]. The main reason is that Sila-Max could spend very short time to enter the stable period of fermentation to produce a large amount of lactic acid to rapidly reduce the pH and inhibit the production of BA and ammonia-N [26, 38].Therefore, lactic acid bacteria additives Sila-Max could improve the silage fermentation qualities of triticale, rye, and oat silages in Qinghai-Tibetan Plateau. But for Sila-Mix, it elevated the pH value and PA content, reduced the WSC content, and did not have any influence on the contents of LA and AA in triticale, rye, and oat silages, which is in accordance with Chen et al. (2021) showing that inoculation of exogenous lactic acid bacteria exerted a limited influence on the silage fermentation and bacterial community compositions of reed canary grass straw on the Qinghai-Tibetan Plateau [49].
4.3 Triticale variety ‘Gannong No.2’is the best raw material to produce quality silage in Qinghai-Tibet alpine area
DMY is the weight while the DM content of the forage is 100%, which plays an important role for silage production [50]. High DMY means high economic benefits [51]. The DMY of triticale variety ‘Gannong No.2’ cutting at the milky stage (14.11 t·hm-2) was significantly higher than that of rye and oat ( Figure 1(a)) due to its thick stem (4.25 mm) and blade (317.50 μm), abundant tillers (5-6/plant), large size of leaves (20.34 cm2) [52]. In contrast, although the plant height of rye variety ‘Gannong No.1’ is the highest, but because of its slender stems (3.21 mm), small size of blade (15.22 cm2) and thin leaves (243.00 μm) [52], it obtained low DMY. And for oat, the lowest height decided its low DMY [53]. The DMY results obtained from this experiment was in consistent with the results of Zhao et al (2018), Wang et al. (2020) [26, 54]. This demonstrated that triticale variety ‘Gannong No.2’ is much more economic than that of oat and rye to make quality silage in Qinghai-Tibet Plateau.
WSC is the most important energy substance for maintaining fermentation quality of silage andhigh content of WSC could provide sufficient energy substances for silage [55]. The WSC in the silage should be at least 1% ~ 1.5% of the fresh weight to ensure successful fermentation [56]. In this experiment, the WSC content of triticale silage produced at the milky stage was 13.73% (with no additive), indicating that triticale itself contained higher WSC. Researchers also mixed triticale in some leguminous silage to elevate the WSC content of silage [57]. As an important volatile fatty acid in silage, PA possesses antifungal properties and it plays an important role in inhibiting aerobic spoilage of silage to preserve the nutritional content of silage [58-59].In oat silage, PA effectively inhibited the formation of undesirable microorganisms and BA, enhanced the aerobic stability and improved the fermentation quality [60]. In this study, PA contentin triticale silage was about 10-20 times higher than that of rye and oat, showing that triticale silage had good aerobic stability, which can be preserved longer than that of rye and oat. Besides, pH value and NH3-N/TN content was also low in triticale silage. Therefore, triticale variety ‘Gannong No.2’ was the best raw material to produce quality silage in Qinghai-Tibet alpine area.”
